# Developmental Transcriptomic Analysis of the Cave-Dwelling Crustacean, *Asellus aquaticus*

**DOI:** 10.3390/genes11010042

**Published:** 2019-12-29

**Authors:** Joshua B. Gross, Dennis A. Sun, Brian M. Carlson, Sivan Brodo-Abo, Meredith E. Protas

**Affiliations:** 1Department of Biological Sciences, University of Cincinnati, Cincinnati, OH 45221, USA; grossja@ucmail.uc.edu; 2Department of Molecular and Cell Biology, University of California at Berkeley, Berkeley, CA 94720, USA; dennis.a.sun@berkeley.edu; 3Department of Biology, College of Wooster, Wooster, OH 44691, USA; bcarlson@wooster.edu; 4Department of Natural Sciences and Mathematics, Dominican University of California, San Rafael, CA 94901, USA; sivan.brodo-abo@students.dominican.edu

**Keywords:** regressive evolution, de novo transcriptome, differential expression, troglomorphy, cave

## Abstract

Cave animals are a fascinating group of species often demonstrating characteristics including reduced eyes and pigmentation, metabolic efficiency, and enhanced sensory systems. *Asellus aquaticus*, an isopod crustacean, is an emerging model for cave biology. Cave and surface forms of this species differ in many characteristics, including eye size, pigmentation, and antennal length. Existing resources for this species include a linkage map, mapped regions responsible for eye and pigmentation traits, sequenced adult transcriptomes, and comparative embryological descriptions of the surface and cave forms. Our ultimate goal is to identify genes and mutations responsible for the differences between the cave and surface forms. To advance this goal, we decided to use a transcriptomic approach. Because many of these changes first appear during embryonic development, we sequenced embryonic transcriptomes of cave, surface, and hybrid individuals at the stage when eyes and pigment become evident in the surface form. We generated a cave, a surface, a hybrid, and an integrated transcriptome to identify differentially expressed genes in the cave and surface forms. Additionally, we identified genes with allele-specific expression in hybrid individuals. These embryonic transcriptomes are an important resource to assist in our ultimate goal of determining the genetic underpinnings of the divergence between the cave and surface forms.

## 1. Introduction

Cave animals are fascinating organisms that frequently share a common suite of characteristics, including reduced eyes, reduced pigmentation, metabolic differences, and enhanced sensory systems. Questions that have long fascinated cave biologists include how and why these characteristics have evolved and whether the same underlying mechanisms mediate trait loss between different cave populations and different cave species.

Historically, it has been challenging to understand how and why cave characteristics have evolved, due to difficulties with rearing cave organisms in captivity and a lack of contemporary experimental resources (e.g., genomic, genetic, and functional molecular tools) for most cave species. In recent years, however, there have been vast expansions of, and improvements in, resources and tools for emerging model organisms. Obtaining genomic information is now possible for most systems, and the complete genomic sequence is available for a limited number of cave-dwelling species [1,2]. In addition, many studies have involved transcriptome sequencing projects for cave dwellers such as crayfish, salamanders, amphipods, isopods, and fish [3,4,5,6,7,8]. For the vast majority of these projects, adult samples have been utilized due to the challenge of obtaining embryonic samples of natural, cave-dwelling species. However, for several cave species, many trait differences are established early in embryonic development, underscoring the importance (and value) of analyzing gene expression differences across embryonic development.

The most widely studied cave model species is *Astyanax mexicanus*, where it is possible to work with embryos and obtain embryonic samples, as well as perform genetic analyses (reviewed in [9,10]). Both adult and embryonic transcriptomes, as well as a draft genome sequence for cave and surface morphs [2,11,12], have been generated. Additionally, contemporary genomic tools, such as gene editing, provide the ability to functionally analyze candidate genes discovered through transcriptome sequencing [13,14,15]. Because of the wealth of data provided by these emerging resources, historical questions impacting on the evolution of cave animals can now be addressed (reviewed in [16,17]).

Despite the great deal of information provided by decades of research in *A. mexicanus*, studies from additional cave organisms are necessary to understand the convergence of regressive loss across animals that inhabit the cave biome. Specifically, the mechanisms that mediate regressive loss in *A. mexicanus* may differ from those mechanisms operating in other cave-adapted species. Thus, it is important to develop other species in a similar way to *A. mexicanus* in order to widen our perspective and to gain a broader understanding of how cave evolution occurs across diverse taxa.

Unfortunately, not every cave animal is amenable to develop as a model in the same way as *A. mexicanus*. There are many considerations, foremost of which is the ability to raise and breed a species in the lab. This feature greatly reduces the number of cave-adapted species for which genetic and developmental studies are feasible. Another important feature for these investigations is an extant surface-dwelling form capable of interbreeding with cave morphs. Owing to the divergence times between cave and surface morphs, the ability to produce viable hybrid offspring is very unusual among studied cave organisms.

*Asellus aquaticus* is a freshwater crustacean that has two morphs, a cave and surface form, both of which can be raised in the lab and can interbreed [18]. Much of the historical work on *Asellus aquaticus* has included comparative morphology between the surface and cave forms, and population genetic analyses of several cave and surface populations throughout Europe [19,20,21,22,23,24]. Recently, a classical genetics approach has been made possible by multiple crossing strategies to create F1, F2, and backcross pedigrees between cave and surface populations. These studies have resulted in production of a linkage map, insight into the genetic architecture of this species, and identification of genomic regions associated with different cave-associated phenotypes [25,26,27].

Though advances have been made in genomic mapping alongside the development of genetic resources, the identity of genes responsible for these trait differences between cave and surface forms remains unknown. A powerful approach to identifying genetic differences between cave-and surface-dwelling forms is comparative transcriptomics. Transcriptomes have been characterized for the Pivka Channel of Planina Cave population, the Molnár János Cave population, and nearby surface populations to both caves [6,28]. Though these studies have been useful in generating genetic resources, the causative genes mediating differences between cave and surface populations have not been established. Part of the issue, as discussed above, is that adult samples are not the most appropriate, as many different characteristics between cave and surface individuals are established during embryonic development [27,29]. For example, eye loss and pigment loss are established by the end of embryogenesis. To investigate the genetic pathways responsible for eye and pigment loss, the most appropriate samples to sequence would be those obtained at this timepoint in embryonic development.

To address this gap in knowledge, we generated de novo embryonic transcriptomes from one cave and one surface population, as well as from hybrid individuals. We hypothesized that many genes would be differentially expressed between cave and surface forms, including those involved in neurogenesis, pigment development, eye development, and metabolism. Furthermore, we expected that a subset of these differentially expressed genes would also show allele-specific expression, suggesting that regulatory mutations result in altered transcriptional abundance for those genes.

## 2. Materials and Methods

### 2.1. Animals

Animals were collected from Rakov Škocjan (surface) and the Rak Channel of Planina Cave (cave) (Figure 1A). Animals were reared in water, lighting, and food conditions as previously described [25,26,27]. Briefly, the animals were kept in an incubator at 12 °C with no lights and were only exposed to light when they were removed from the incubator or the incubator door was opened. Surface animals were raised in tanks with around 10 individuals per tank. Similarly, cave animals were raised in tanks with around 10 individuals per tank. Hybrid crosses were generated by mating a single cave male to a single surface female. When a female with embryos was observed in any of the above tanks, the females were monitored until the embryos were around 70% of the way through development. They were then removed from the female using a clove oil solution of 20 µL in 50 mL of fresh water as previously described [27]. Embryos were kept in a small dish with commercial spring water (Crystal Geyser) until they reached 90% of embryonic development, when both pigmentation and incipient ommatidia were present in the surface, but not cave, embryos (Figure 1B,C) [27].

### 2.2. RNA Extraction, Library Preparation, and Sequencing

An entire brood at 90% of embryonic development was used for a single sample, which ranged from 25–89 embryos. Embryos were extracted in 200 µL of TRIzol (Thermofisher, Waltham, MA, USA) and mechanically disrupted using an Eppendorf pestle. Samples were sent to the Functional Genomics Lab, Vincent J. Coates Genomics Sequencing Laboratory, California Institute for Quantitative Biosciences (QB3) University of California, Berkeley. Total RNA was extracted using the TRIzol Thermofisher protocol. PolyA selection was performed, and library preparation was performed using the low input protocol of the Nugen kit. Sequencing was performed using 150 bp paired end reads on both the Illumina HiSeq 4000 and the HiSeq 2500 sequencing machines to ensure a quality dataset by the sequencing facility.

### 2.3. De Novo Transcriptome Assembly and Annotation

A total of nine samples, which produced 36 fastq files, were processed for transcriptome assembly and annotation. We evaluated three *Asellus* cave embryonic samples (MPD1, MPD5, MPD6), three surface embryonic samples (MPD2, MPD3, MPD8), and three hybrid embryonic samples (MPD4, MPD7, MPD9) subjected to pair-read sequencing and processed in duplicate (total = 36 files). To achieve the most accurate mapping for downstream RNA-seq studies, we built morphotype-specific transcriptomes using SeqMan NGen (DNAStar, Madison, WI, USA). Initial de novo assemblies utilized the default assembly parameters for SeqMan NGen, including a mer size of 21, a minimum match percentage of 80%, and a maximum cluster size of 100,000, resulting in incompletely assembled contigs. This was based on the fact that default parameters yielded far fewer transcripts >1 kb in length. We sought to increase the average transcript lengths of our assemblies, and we tested a variety of parameters. We found the optimal results when we adjusted the mer size (19) and increased the minimum match percentage (to 97%) and maximum cluster size (to 300,000). This approach resulted in a 30% increase in the number of transcripts >1 kb in length (from 37,769 to 49,146 in the surface assembly). This approach also provided the longest mean transcript lengths (surface = 1061 bp, cave = 1069 bp, hybrids = 952 bp), as well as the most assembled transcripts >1 kb in length (surface = 49,233, cave = 51,822, hybrids = 52,390; Table 1). Additionally, we performed a BUSCO analysis and found highly similar results across all three transcriptomes, as well as an “integrated transcriptome” that utilized every read we generated (Appendix A). We felt that the longest transcripts represented the best individual transcript assemblies, and therefore proceeded to annotate those assembled transcripts that were 1000 bp or longer.

All annotations were carried out using Blast2GO (v.5.2.5) running Java v.1.8.0_144. To capture the most comprehensive information, we performed two rounds of BLAST-associated annotations for each of three transcriptomes—one using the *Tribolium castaneum* genome as a reference and one using the SwissProt database (Table 2). The latest SwissProt database was downloaded from within Blast2GO, using the following link: https://ftp.ncbi.nlm.nih.gov/blast/db. The *Tribolium* database used for our blast-based annotation was downloaded from ftp.ncbi.nlm.nih.gov. We used the *Tribolium* reference that was uploaded in March 2016. Although the SwissProt database is perpetually being updated, both databases were retrieved (and all annotation tasks were performed) in May 2018. In brief, we submitted a fasta-formatted file containing all de novo-assembled sequences to Blast2GO, specified our database of interest, and proceeded through all default annotation steps. We implemented a script to remove all annotated transcripts associated with ribosomal or mitochondrial sequences, which ranged between 734–1066 sequences with an identified blast hit. For all three transcriptomes (surface, cave, and hybrids), we obtained comparable results for both databases; however, the *Tribolium castaneum* reference provided the largest number of successful annotations.

### 2.4. RNA-Sequencing and Expression Analyses

Once annotation was completed using Blast2GO, we performed RNA-sequencing analyses using ArrayStar (v.13; DNAStar, Madison, WI, USA). We performed RNA-seq analyses for all transcriptome references (i.e., *Tribolium castaneum* and SwissProt). Specifically, we mapped reads from all of our embryonic samples to all of our transcriptome references. All of our analyses retrieved very similar results. Our workflow involved mapping sequencing reads from all three morphotypes (cave, surface, and hybrids). Gene expression results were normalized using reads per kilobase per million mapped (RPKM). This normalization strategy is necessary to control for differences in sequencing depth between samples and to compare expression levels of transcripts that differ in length. Our resulting dataset included a measure of linear total RPKM, which provides a statistical metric of expression that could be compared across datasets. This metric was then used to compare expression (based on fold change difference) between groups (e.g., cave versus surface).

We compared assemblies of different transcriptome reference files to evaluate the consistency of this calculated expression metric and found them to be highly similar. However, our process of annotation using Blast2Go periodically yielded more than one blast hit to a single, orthologous reference transcript. To deal with this issue, we averaged the expression values (i.e., RPKM values) for contigs that blasted to the same reference transcript. This yielded the most accurate expression value for each annotated gene. This calculation enabled us to correct for multiple blast hits to the same reference, however it may have inadvertently collapsed the expression for different isoforms (or paralogues) into a single transcript. This project could not evaluate the possibility of *Asellus aquaticus*-specific isoforms or paralogous genes, a caveat that will need to be addressed in future genome sequencing projects. Finally, given the inaccessibility of fresh tissues (with which to extract RNA for quantitative PCR validation), we used a variety of filters to maximize the validity of our reported differentially-expressed genes.

### 2.5. Allele-Specific Expression Using ASE-TIGAR

To assess allele-specific expression of differentially expressed genes, pairs of transcripts were identified across cave and surface transcriptomes if they had the same *Tribolium castaneum* Uniprot ID. For a given pair of alleles, transcripts were manually trimmed to be similar in length, based on sequence identity (Figure 2B; Appendix A). We then used the ASE-TIGAR software [30] to generate transcript abundances for each allele. The software was supplied a single FASTA file containing both trimmed alleles from the cave and surface transcriptomes, as well as paired-end reads from the MPD4, MPD7, and MPD9 hybrid embryo transcriptomes. The output of this software was a file containing the expected number of fragments mapped by ASE-TIGAR, an FPKM value, and a THETA value, which was the estimated transcript abundance. We used this THETA value as our metric of expression for each allele. Given that the list of genes we selected for allele-specific expression analysis could be biased towards genes that might show allele-specific expression, we determined that it was important to have a statistically rigorous approach to identifying genes with true allele-specific expression differences. An ideal null distribution for hypothesis testing in this scenario would be the distribution of all log fold change values for all pairs of genes. However, generating such a dataset was neither practical nor computationally feasible. Instead, we chose to simulate a null distribution that represented the intra-allele variance using the THETA values calculated for each allele in each replicate (MPD4, MPD7, MPD9). This null distribution would convolve noise arising from technical differences (batch effects, sequencing errors, etc.) and biological differences (gene expression variability between samples, gene expression noise, etc.). We generated an intra-allele null distribution by comparing inter-replicate log fold changes for all replicates within a given allele, e.g., gene X, surface allele replicate 1 vs. gene X, surface allele replicate 2, etc., using a custom Python script. Genes that had THETA = 0 in one or more replicates of one or more alleles were filtered out of the analysis. We then compared the distributions of intra-allele variations for surface alleles and cave alleles using a two-sample Kolmogorov–Smirnov (K–S) test and found that the two distributions were indistinguishable (K–S statistic = 0.0289, *p*-value = 0.9643). We merged the surface and cave allele null distributions and used this total distribution as a null distribution for assessing significance, also using a two-sample Kolmogorov–Smirnov (K–S) test. For each pair of alleles, we generated a distribution of log fold changes by comparing each replicate of one allele to each replicate of the other allele, for a total of nine values per pair of alleles. We used the two-sample K–S test implemented in the Pandas Python package to generate a K–S statistic and a *p*-value, and then performed the Benjamini–Hochberg (B–H) multiple hypothesis testing correction procedure to that *p*-value using a Scipy.stats Python package and α = 0.05. Genes for which significant log fold change differences were observed based on this B–H corrected *p*-value were called genes with true allele-specific expression differences.

### 2.6. Data Deposition

All sequences analyzed in this report have been provisionally submitted to the National Center for Biotechnology Information, Sequencing Reads Archive (BioProject ID:PRJNA597080).

## 3. Results

### 3.1. Characterization of Surface, Cave, and Hybrid Transcriptomes

Following the optimization of our assembly parameters, we retrieved highly similar results for all three of our assembled transcriptomes (Table 1; Appendix A). The total number of reads that were assessed for each transcriptome was very similar between surface morphs (~364 M), cave morphs (~361 M), and hybrids (~394 M). The total number of assembled reads for surface (~155 M), cave (~164 M), and hybrids (~132 M) were similarly comparable, although a higher proportion of assembled reads were utilized in cave morphs (45.5%) compared to surface morphs (42.4%) and hybrids (33.5%). The reduced proportion of assembled reads used in the hybrid transcriptome assembly may reflect the sequence divergence between cave and surface morphs.

All assemblies were completed in roughly the same amount of time (50.4–54.2 h) and yielded comparable numbers of transcripts (surface = 113 K; cave = 119 K; hybrid = 143 K), or comparable average lengths (surface = 1061 bp; cave = 1069 bp; hybrid = 952 bp). Our goal, however, was to annotate the best-characterized transcripts in each dataset. We reasoned that the longest transcripts represented the best individual transcript assemblies, and therefore proceeded to annotate those assembled transcripts that were 1000 bp or longer. This value was similar across all three assemblies: surface = 49,233; cave = 51,822; hybrids = 52,390 (Table 2).

Using these assemblies as a starting point, we subjected each transcriptome to comprehensive annotation using Blast2GO (Methods). This BLAST-based approach was performed against the *Tribolium castaneum* genome and SwissProt database, in order to compare the quality of each database. We chose these databases because *Tribolium castaneum* is an arthropod with a comprehensive genome database, and the SwissProt database is an open-access and manually annotated resource of protein sequence and functional information. Overall, we found that the average percentage of failed BLAST hits was higher when we used the SwissProt database (mean = 61.3%) compared to the *Tribolium castaneum* database (mean = 58.4%). Consequently, our final transcriptome size was larger when we annotated against the *Tribolium* (mean = 19,727 transcripts) compared to the SwissProt database (mean = 18,110 transcripts). In sum, our results indicated that the *Tribolium castaneum* database provided better results (Table 2), and therefore our downstream analyses utilized these annotated transcriptomes.

### 3.2. Differential RNA-Seq Analysis Between Cave and Surface Morphs

We mapped the cave and surface reads separately to each of the four different transcriptomes: cave, surface, hybrid, and integrated transcriptomes. We selected all genes that had at least a two-fold change in the same direction (increased or decreased expression) between cave and surface in all four experiments and had a standard deviation of less than or equal to 8. Then we selected the top 50 genes that were underexpressed in the cave form and the top 50 genes that were overexpressed in the cave form to analyze further (Figure 1E; Appendix A).

Several of the genes that were underexpressed in the cave made biological sense, as they are involved in eye or pigment function such as *long-wavelength sensitive opsin*, *cell cycle control protein 50A-like*, *membrane-bound transcription factor site 1 protease-like protein, scarlet-like protein, protein pygopus-like*, and *atonal*. Genes that were overexpressed in the cave form include those involved in metabolism, such as *solute carrier family 35 member F6-like protein*, *gamma-glutamyltransferase 7-like protein*, and *inositol oxygenase-like protein*. Also overexpressed in the cave samples was *annulin-like protein*, which is expressed in stripes in each limb bud segment [31] and could be a candidate for differential antennal characteristics in the cave form.

### 3.3. Allele-Specific Expression Analysis in Hybrid Individuals

Genes that display differential expression between populations may arrive at this difference through both cis- and trans-regulatory mechanisms. In cis-regulatory changes to gene expression, a change to the DNA sequence either within a gene or in regulatory elements thereof is responsible for an observed expression difference between populations (Figure 2A). When trans-regulatory factors change gene expression, the regulatory sequence of a gene may not change, but instead, a change to the expression of a trans-regulatory factor (an activator, repressor, etc.) between populations drives the difference in expression of a downstream gene. By examining the expression of alleles of a given gene in hybrid organisms, one can determine mechanisms of gene expression difference, whether they be cis-regulatory, trans-regulatory, or a combination of both. In hybrid animals, trans-regulatory effects are normalized across alleles, as both alleles existing in the same nucleus are subjected to the same input by activators and repressors. As such, when expression differences in alleles are observed in hybrids, one possible explanation is that cis-regulatory changes contribute to differential expression between populations (Figure 2A). Allele-specific differences can also come about due to parent-of-origin effects (see Discussion).

We wanted to examine the mechanism of differential gene expression for the genes we identified as differentially expressed between cave and surface populations. To do this, we performed allele-specific expression (ASE) analysis using the ASE-TIGAR software package [30]. This software, given a FASTA file containing both isoforms of a gene and FASTQ reads from hybrid animals, generates transcript abundance estimates for each allele (Figure 2B). We identified pairs of alleles for the most differentially expressed genes and generated a log fold change value for the usage of surface vs. cave alleles in hybrid animals (Figure 2). We then used a two-sample Kolmogorov–Smirnov test with a Benjamini–Hochberg multiple hypothesis testing correction to call significance of observed ASE, using intra-allele log fold change as our null distribution (see Methods; Figure 2C,D). Overall, genes with significant ASE tended to have larger log2 fold change between the two alleles (Figure 2E; Appendix A).

Many of the genes we identified as highly differentially expressed (DE) between individuals of different populations also appeared to show ASE between alleles in hybrid animals (Figure 2E, Figure 3A,B; Appendix A). For example, the long-wavelength sensitive opsin gene was found to be about four-fold (mean log2 fold change across transcriptomes) underexpressed in cave than surface animals (mean = 4.15 (log2 scale), SEM = 0.33), and was the most surface-biased gene by DE analysis. In hybrid animals containing one surface and one cave allele, we observed that the same gene showed a 10-fold (mean log2 fold change) difference between alleles (mean = 10.497, SEM = 2.79). Cis-regulatory changes may contribute to differences in *long-wavelength sensitive opsin* expression between populations.

By examining all genes with significant ASE, we observed that most of the genes likely had some cis-regulatory component to their change in expression between populations. We inferred this result because genes that showed DE in favor of surface animals, on the whole, tended to also show ASE in favor of the surface allele (23 out of 26 genes, Figure 3A,B). Meanwhile, genes that showed DE in favor of cave animals also tended to have ASE in favor of the cave allele (16 out of 19 genes, Figure 3A,B). For six genes (Figure 3B, marked with asterisks), we observed significant ASE that showed a strong bias in the opposite direction from what we expected from the DE analysis. For example, *C-terminal binding protein-like protein* and *maltase A1-like protein* were found to be more highly expressed in surface animals, but by ASE the cave allele appeared to be more expressed. Such results can be explained through models of competing cis-by-trans effects.

## 4. Discussion

### 4.1. Candidate Genes

Typical features of cave animals include loss of eyes, loss of pigment, differences in metabolism, and enhanced sensory structures. Specifically in *Asellus aquaticus*, the cave form can show loss of eyes, loss of pigment, and increased appendage length [22,23,32]. Less is known about metabolic and behavioral differences between the cave and surface populations, but a recent study showed that acetylcholinesterase and glutathionine S transferase had lower activity in cave individuals as compared to the surface individuals, supporting the idea that the cave form has lower metabolic and locomotor activity [33]. Differences in allele-specific expression were also seen in *glutathione S transferase mu 5* from a previous analysis [6]. In addition, shelter-seeking behavior has been shown to be different between some cave and surface populations [34]. Overall, we expected to find differential expression and allele-specific expression in genes involved in eye development, pigmentation, appendage development, and metabolism. As expected, some of the differentially expressed genes that we found to be differentially expressed have been shown to play a role in phototransduction, photoreceptor development, and/or eye development, such as *atonal*, *long-wavelength sensitive opsin*, *cell cycle control protein 50A-like*, *membrane-bound transcription factor site 1 protease-like protein*, *protein EFR3 homolog cmp44E-like protein*, *pygopus-like protein*, and *domeless*. Furthermore, a subset of the above (*long-wavelength sensitive opsin*, *cell cycle control protein 50A-like*, *membrane-bound transcription factor site 1 protease-like protein*, *pygopus-like protein*, and *protein EFR3 homolog cmp44E-like protein)* also showed allele-specific expression indicating that cis-regulatory changes may be responsible for the differential expression of those genes. Fewer genes with known involvement in pigmentation were observed. *Scarlet*, a gene involved in pigment transport [35], was overexpressed in the surface form; however, *scarlet* was not shown to have allele-specific expression and therefore is unlikely to have a cis-regulatory change. *Annulin-like protein* was another gene of interest that was overexpressed in the cave form as compared to the surface form and had higher cave allele expression in the hybrids. Interestingly, this gene is expressed in grasshoppers in stripes along the forming limb segments and could be a candidate for appendage length changes in *A. aquaticus* [31]. Another gene of interest that had showed higher ASE for the cave allele was *myotubularin-related protein 9-like protein* (*MTMR9*). Polymorphisms in this gene have been shown to be associated with obesity and glucose tolerance in Genome-wide association studies (GWAS) in humans [36,37]. One study in *Asellus aquaticus* found that the surface form had a greater feeding activity than the cave form [38], but little else is known regarding differences in food acquisition in the cave environment for *Asellus aquaticus*. However, studies in the cavefish *Astyanax mexicanus* have shown that some cave populations are insulin resistant and able to binge eat [39,40].

Another interesting gene that showed both expression differences between populations and allele-specific differences is *gamma-glutamyl transferase 7-like protein* (*GGT7*). Elevated GGT is commonly seen in individuals with non-alcoholic fatty liver disease [41]. Interestingly, one of the cave populations of the cavefish *Astyanax mexicanus* develops fatty livers when exposed to high-nutrient conditions [39]. Lipid, carbohydrate, and protein amount have been examined in wild-caught cave and surface specimens of *Asellus aquaticus*, and little difference was observed [42]. Future studies can examine whether lab-reared cave and surface forms of *Asellus aquaticus* differ in fat storage, insulin resistance, and starvation resistance similar to cave and surface populations of *Astyanax mexicanus*.

### 4.2. Involvement of Regulatory Mutation Versus Coding Mutation in Evolution of Cave Traits

When working with species with limited genomic and genetic resources, most studies that discover the causative genes for particular phenotypes involve coding mutations. This may be due to ascertainment bias, as coding mutations are much easier to identify than cis-regulatory mutations, which could be in much larger (and uncharacterized) regions of the genome. Furthermore, cis-regulatory changes can be more difficult to test functionally than coding mutations. Because of these challenges, most of the mutations and genes identified as causative for cave-related traits in the model system of *Astyanax mexicanus* have been coding mutations [39,40,43,44,45,46,47] though there are some exceptions [48]. Allele-specific expression studies in hybrids are a powerful way of identifying cis-regulatory differences. Here, we have identified many genes with allele-specific expression, some of which likely have cis-regulatory changes, as inferred through a positive correlation between allele-specific expression and differential expression. Studies have indicated that much of evolutionary change occurs via cis-regulatory mutations (reviewed in [49]), and therefore the establishment of techniques to identify such changes in species that have limited genomic and genetic resources is crucial for identifying the genetic/genomic substrate of evolutionary change.

### 4.3. Cis-Versus Trans-Regulation

Another major question in evolutionary biology regards whether cis- or trans-regulatory changes dominate in driving evolutionary change. In trans-regulatory changes, modifications to the expression or function of trans-regulatory factors, such as transcription factors, have a cascading effect on the expression of many other downstream target genes, driving evolutionary changes. Cis-regulatory changes, on the other hand, are more restricted, tend to occur in regulatory regions, and affect the expression of a particular gene. Trans-regulatory change might be expected to cause more drastic and pleiotropic effects, whereas cis-regulatory change would be less likely to have pleiotropic consequences (reviewed in [50]). Previous studies comparing species and interspecific hybrids have shown input of both trans-and cis-regulatory change (reviewed in [50]).

In our study, we have observed possible cases of both cis- and trans- regulation between *Asellus aquaticus* populations. An example of a likely cis-regulatory change is in the case of the *long-wavelength sensitive opsin* gene, for which cave samples showed lower expression than surface samples. In hybrid samples, the cave allele also showed significantly lower expression as compared to the surface allele; the shared directionality of the DE and ASE results for this gene suggests that cis-regulatory effects are responsible for expression differences between the populations. On the other hand, the *scarlet* gene is a likely example of a trans-regulatory change in our dataset. Here, though the cave samples showed lower expression as compared to the surface samples, in hybrid samples, the cave allele was not significantly reduced in expression compared to the surface allele. When both *scarlet* alleles were placed in an identical trans-regulatory environment, the alleles expressed at indistinguishable levels, suggesting that differences in a trans-regulatory factor between the populations is responsible for expression differences. However, we cannot exclude cis-regulation for this gene as it is possible that certain genes show allele-specific expression only in specific tissues and sequencing transcriptomes of entire bodies dilutes out any tissue-specific allele-specific expression [51]. In addition, we observed cases wherein cave samples showed lower expression compared to surface samples, but in hybrids the cave allele had higher expression. This might indicate both trans and cis modes of regulation, which may be evidence for compensatory mutations (reviewed in [50]). However, a recent study indicated that cases of compensatory cis-and trans-regulation are often overestimated as a result of correlated errors that occur when estimating ASE [52]. Our data were not amenable to the analysis presented in this paper, due to the methods we used to estimate ASE. The interplay between cis-and trans-regulation can ideally be examined by comparing the fold change of DE versus ASE. If the fold change of ASE is equal to the fold change of DE, cis-regulation likely explains the differential expression fully [50]. If the fold change of ASE is less than the fold change of DE, a combination of cis-regulation and trans-regulation likely explains the differential expression. Our DE and ASE analyses used different measures of transcript abundance, and therefore the fold changes of each are not directly comparable. Future analyses with greater sample sizes and different measures of transcript abundance may enable finer examination of the differences in ASE and DE for other genes.

### 4.4. Parent-of-Origin Effects Versus Cis-Regulation

Allele-specific expression in hybrid organisms can result from cis-regulatory change or because of parent-of-origin effects, in which the maternal and paternal copies of the gene are expressed differently, as has been observed in different organisms, including mammals, insects, and plants [53]. We cannot exclude parent-of-origin effects in the genes we found to have significant allele-specific expression, but it is likely that many of these genes have cis-regulatory changes. Future studies can eliminate potential parent-of-origin effects by generating hybrid samples from both cave female/surface male and surface female/cave male matings. As the former crosses are considerably more difficult to generate, our study was restricted to samples from the latter type of cross. Future investigations may tease apart the genes that are truly expressed as a result of cis-regulatory changes versus those with parent-of-origin effects, once it is more tractable to generate crosses with cave female and surface male animals.

### 4.5. Comparison to Adult Asellus Transcriptome

There are two previously published transcriptomes from *Asellus aquaticus*, both on mostly adult samples [6,28,54]. The first transcriptome utilized Roche/454 sequencing technology and was more limited in terms of actual sequence generated, though some surface embryonic samples were sequenced [6,54]. This transcriptome was generated from individuals from the Pivka Channel of Planina Cave and a nearby surface population, Planina Polje, both found in Slovenia. More recently, a transcriptome was generated from Hungarian populations of *Asellus aquaticus*, including the Molnár Janós Cave population. This study found that genes involved in phototransduction were still expressed in this cave population [28]. The authors found two expressed opsins, and neither seemed to have drastic coding changes. Consequently, they hypothesized that if vision loss has occurred in this population, it is likely due to the decreased expression of opsins. This idea is supported by our study, which uses a different cave population. Specifically, we found both differential and allele-specific expression in *long-wavelength sensitive opsin*. Our studies have expanded the transcriptomic resources for this species by generating a transcriptome for an additional cave population, the Rak Channel of Planina Cave. This is a useful cave population to examine as comparative embryology, as well as genetic mapping studies, have both been performed for this cave population [25,27]. In addition, this is the first study generating an embryonic transcriptome of a cave population of *Asellus aquaticus* and examining differential and allele-specific expression between cave and surface embryonic samples, giving us a window into the developmental mechanisms resulting in population-specific differences.

### 4.6. Comparison to Other Cave-Dwelling Animal Transcriptomes

Transcriptomes of many cave-dwelling organisms have now been sequenced. Examples include other populations of *Asellus aquaticus*, *Gammarus minus* (an amphipod crustacean) [55], *Niphargus hrabei* (another amphipod crustacean; [28]), cave crayfishes [7,8], *Poecilia mexicana* [5], *Sinocycloheilus* species [4,56], multiple species of cave beetles [57,58], multiple isopod species [59], and *Astyanax mexicanus* [11,12]. Transcriptome studies of these cave animals often look to see whether genes involved in phototransduction are still expressed and whether there are any obvious mutations in genes involved with vision. The majority of the transcriptomes described above are from adult samples, owing to difficulties with breeding or otherwise obtaining embryonic samples. However, embryonic samples have been examined in *Astyanax mexicanus* [6,12].

One approach that has been lacking in studies of cave transcriptomes is using hybrid transcriptomes to evaluate allele-specific expression. A previous study in *Asellus aquaticus* examined allele-specific expression in a limited number of genes from a single adult hybrid sample [6]. In most other cave-dwelling animals, it is not possible to examine allele-specific expression because it requires both a cave and surface form, and they must be capable of interbreeding. However, here we show that where this approach is possible, it is a powerful way to investigate genes that might have cis-regulatory mutations. In the future, this approach can be applied to other species that have surface and cave forms, even those that might not have fertile hybrids or viable hybrids (as long as the hybrids can start development). Potential species to examine include *Poecilia mexicana* and *Gammarus minus*.

### 4.7. Future Steps

Here, we examined comparative expression and allele-specific expression in whole bodies of groups of individuals at a particular developmental timepoint. In the future, we aim to expand our analysis to additional timepoints and potentially specific tissues, as these two factors are known to influence both comparative and allele-specific expression [51]. Additionally, now that methods are established to investigate differential expression and allele-specific expression in embryonic samples of cave versus surface morphs of *Asellus aquaticus*, one next step is to expand the analysis to other cave populations. One of the advantages of working with this species is the number of populations that are thought to be independently evolved [24,60]. By examining gene expression and allele expression differences in these different cave populations, it should be possible to better understand how these cave-specific traits have evolved and determine if the independently evolved populations have evolved similarly or differently. Furthermore, now that a number of candidates with putative cis-regulatory changes have been identified, we can investigate them by placing them to the linkage map to determine if they coincide with mapped regions responsible for eye and pigment variation. Future work developing functional methods in *Asellus aquaticus*, such as genome editing and gene expression visualization, will enable testing of these genes to validate whether they are causative for associated cave-related traits.

## Figures and Tables

**Figure 1 genes-11-00042-f001:**
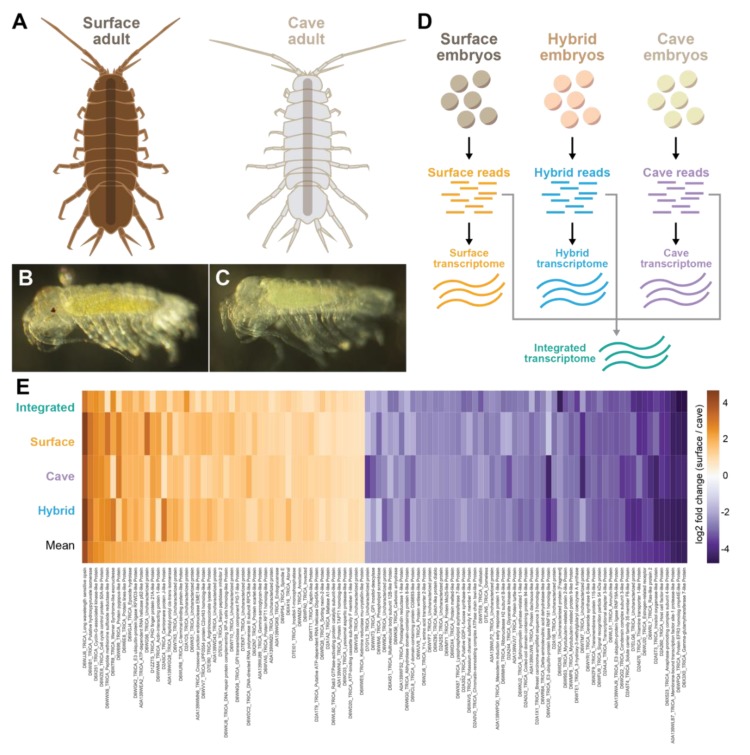
Top 50 overexpressed genes and top 50 underexpressed genes in the cave samples as compared to the surface samples. (**A**) Illustrations of a surface adult and a cave adult. Cave adults lack pigmentation and eyes, and have longer limbs. (**B**) Surface embryo that has gone through 90% of embryonic development. (**C**) Cave embryo that has gone through 90% of embryonic development. (**D**) Four different transcriptomes were generated, one from the cave embryonic samples, one from surface embryonic samples, one from hybrid embryonic samples, and one from all embryonic samples (referred to as the integrated transcriptome). (**E**) Heatmap showing the top 50 downregulated genes in the cave form (various shades of orange) or top 50 upregulated genes in the cave form (various shades of purple). All genes shown had the same direction of fold change and a standard deviation of less than 8 across all four analyses. Uniprot ID and gene name from the *Tribolium castaneum* genome is shown.

**Figure 2 genes-11-00042-f002:**
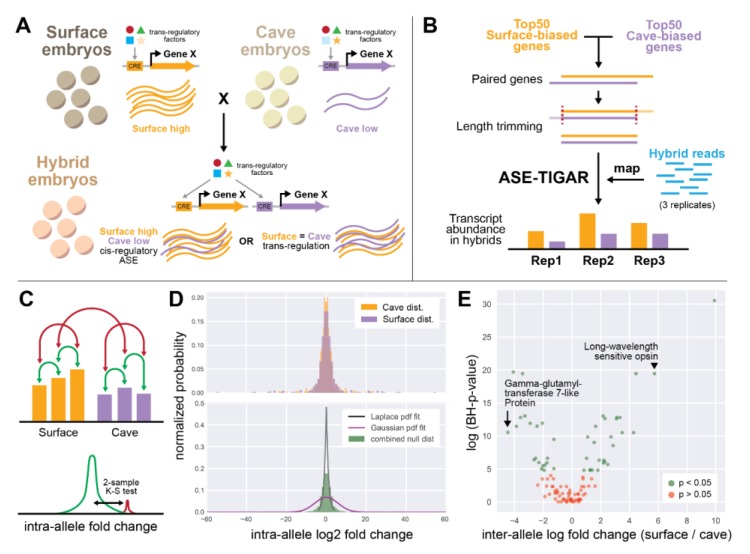
Experimental design of allele-specific expression analysis. (**A**) A hypothetical example of gene X, which is differentially expressed between cave versus surface individuals and also shows allele-specific expression with lower expression of the cave allele. In this case, the underlying mechanism may be a result of differences between trans-regulatory factors between populations, or in the cis-regulatory sequence. If cis-regulatory mechanisms dominate, then we expect to see similar surface-biased expression in hybrids. However, if trans-regulatory mechanisms dominate, then the equalized trans-regulatory environment in hybrids will result in no allele-specific expression. (**B**) Pipeline of allele-specific expression (ASE) analysis. The top 50 differentially expressed genes, in both directions, present in both the cave and surface transcriptomes were selected, paired genes were trimmed to the same length, and hybrid reads were mapped to the trimmed cave and surface versions of each gene. (**C**) Intra-allele log2 fold change was calculated by comparing transcript abundance between replicates of a given allele (green arrows). This null distribution (green curve) was then compared to distributions of inter-allele log2 fold change (red arrows, red curve) using a two-sample Kolmogorov–Smirnov test. (**D**) Intra-allele log2 fold change distributions for cave and surface populations. The top panel shows each allele separately, overlapped. These distributions were determined to be indistinguishable (K–S test, see Methods). The bottom panel shows the combined distribution (green), and a Laplace fit (black line) and Gaussian fit (magenta line) to the distribution. The combined null distribution does not neatly fit either a Laplace or Gaussian distribution, validating that the two-sample Kolmogorov–Smirnov test is appropriate, as it does not assume that either distribution is parametric. (**E**) The analysis identified 45 genes that had significant allele-specific expression (green) and 55 genes that did not have significant allele-specific expression.

**Figure 3 genes-11-00042-f003:**
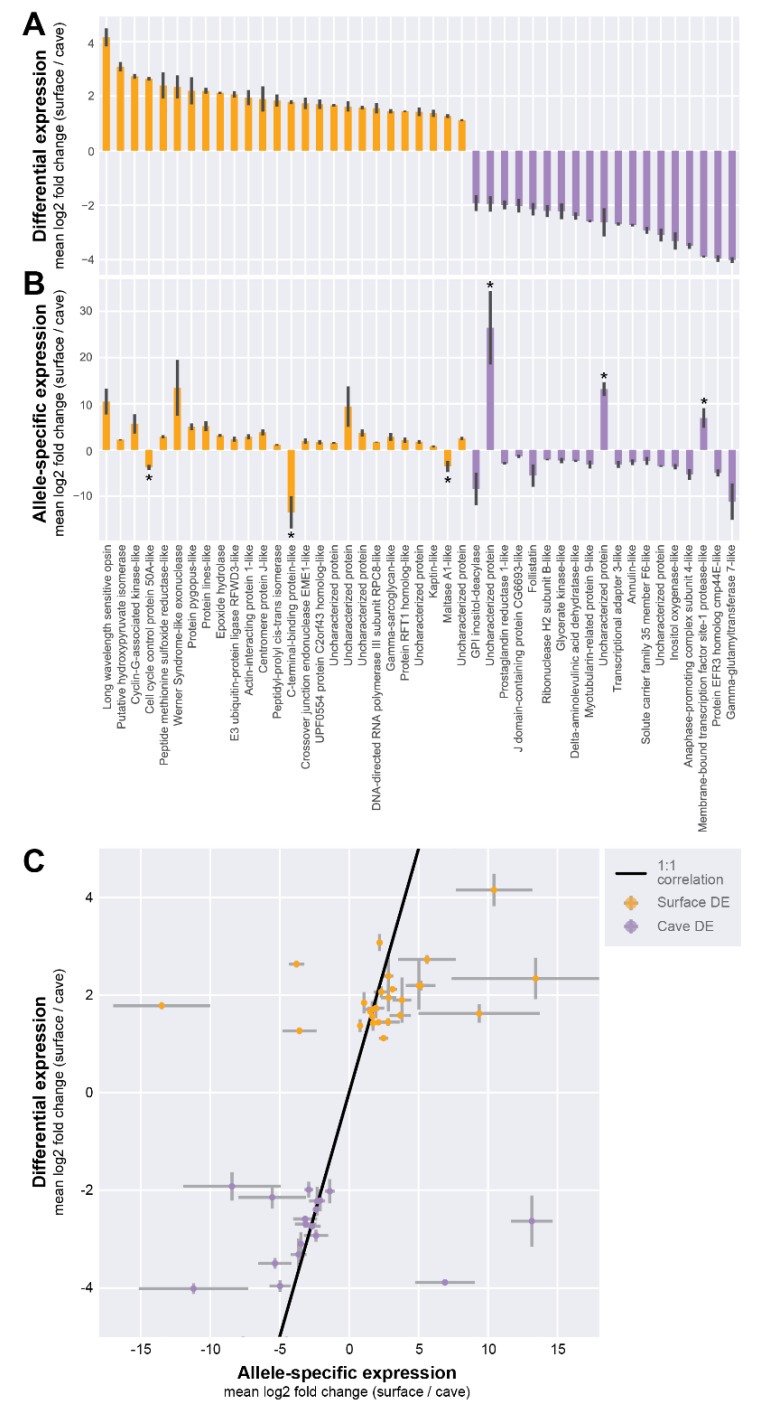
Allele-specific expression suggests cis-regulatory contribution to population difference. (**A**) For the subset of genes that showed significant allele-specific expression, mean log2 fold change comparing surface and cave differential expression. (**B**) Mean log2 fold change comparing surface allele and cave allele expression within the hybrid, in the same order as in (**A**). (**C**) A Spearman correlation test indicates that differentially expressed (DE) and ASE analyses are significantly correlated (correlation = 0.5241, *p*-value = 2.197 × 10^−4^). Note: The gene with the highest allele-specific expression, an uncharacterized protein, was omitted from the scatter plot for ease of visualization.

**Table 1 genes-11-00042-t001:** Comparison of transcriptome assemblies of *Asellus* cave morphs, surface morphs and hybrid individuals.

	Surface Morphs	Cave Morphs	Hybrids
*Sequence Read Summary*			
Total Assembled Reads	155039720	164487662	132336702
Total Unassembled Reads	83386227	109047509	84947885
Total Reads Excluded by Sampling	126373422	87592389	176811792
Total Number of Reads	364799369	361127560	394096379
*Transcript Summary*			
Total number of Transcripts	113432	119569	143962
Average Length of Assembled Transcripts	1061	1069	952
Assembled Transcripts >1kb	49,233	51,822	52,390
*Assembly Time*	50.7 h	54.7 h	54.2 h

**Table 2 genes-11-00042-t002:** Annotation results against two reference databases for Asellus cave morphs, surface morphs and hybrid de novo transcriptomes.

	**Surface Morphs**	**Cave Morphs**	**Hybrids**
*Tribolium Genome Database*			
Total Number of Transcripts	113,432	119,569	143,962
Assembled Transcripts >1kb	49,233	51,822	52,390
No BLAST hits	28,648	30,340	30,709
Ribosomal sequences	518	749	712
Mitochondrial sequences	880	734	973
Total number of annotated sequences	19,187	19,999	19,996
	**Surface Morphs**	**Cave Morphs**	**Hybrids**
*SwissProt Database*			
Total Number of Transcripts	113,432	119,569	143,962
Assembled Transcripts >1kb	49,233	51,822	52,390
No BLAST hits	29,918	31,928	32,157
Ribosomal sequences	603	624	839
Mitochondrial sequences	986	992	1066
Total number of annotated sequences	17,726	18,278	18,328

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
