# Peer review of "Developmental Transcriptomic Analysis of the Cave-Dwelling Crustacean, Asellus aquaticus"

_genes, 2019, doi:10.3390/genes11010042_

Round 1

Reviewer 1 Report

The paper “Developmental transcriptomic analysis of the cave dwelling crustacean, Asellus aquaticus” by Gross et al describes a new assembly for cave, surface, and hybrid crustacean transcriptomes and comparative analysis of its expression an advance that I think would be appropriate for publication in Genes. I have only a few comments that I would like to see addressed prior to publication.

Figure 1A and B showing the feature of two Asellus aquaticus, are they similar size between two habitats? Adding scale bar will be helpful. Figure 1E showing the 50 overexpression and underexpression genes, It showing just genes list, so what about gene ontology? These gene need categorized based on gene ontology, it provide more useful information. The authors sequenced and assembly separated each samples, so orthology analysis between cave, surface, and hybrid based on contigs, not expression, will be helpful for understanding cave Asellus aquaticus feature.

Reviewer 2 Report

The paper from Gross at al., aims to investigate a very interesting issue in identifying differences between cave and surface individuals of Asellus aquaticus. Looking at those genetic differences not in adults but in morph, where most of the “adaptive” changes might happen, make this paper very interesting and worth publishing. Furthermore, the novelty of the paper also consists on using transcriptomics to assess those differences and on performing the allele-specific analysis. The data generated from this study will be of great use for the community.

However, I have major concerns on the methods used. My major issues are related on how the authors assess the differential expression between one dataset and another. They refer to “RNA-Seq analysis” (line 162) which I am not sure what they mean.

Overall, there is a lot of “back and forth” in the MS between methods and results and some key information missing from the methods are in the results. In addition, there are a lot of missing information such as the kit used, software’ version, data of download (see specific description).

In details:

Line 53. The “star” change with a more scientific term (e.g. a good model)

Line 68-99. Most of this could be simplified and eliminated

Line 111. Although it refers to a published paper, it would be good to give a brief description (e.g. “Briefly”) specially referring to light and temperature conditions

Line 123. “embryos” which specific stage the authors refer to?

Line 127. “Trizol protocol” is there a reference? A kit used?

Line 130. I am confused here on why the authors performed the sequencing of the same sample twice on 2 different platforms. If the reason was to increase the coverage, why not have a better depth of sequencing with one platform? Is there a different reason for that?

Line 132. Change d to D

Line 133. A total of 36 samples-The habit of RNASeq is not to mention the PE files as separate. Forward and Reverse files get processes together. Based on this the number of sample is 18. Furthermore, each sample got sequenced twice. Thus I will suggest for a better clarity to mention 9 samples (the biological ones)

Line 138. “initial mapping” what this refer to? Mapping against the transcriptome? Of which files?

Line 140. How the “incompleted assembled sequeneces” are calculated?

Line 143. From what I am understanding here, the authors generated 4 de novo assemblies and are trying to change settings to get the best output. My first concern is that they only are interested here on the longest transcripts as measure of quality. When generating a de novo transcriptome, there are several metrics to evaluate the quality such as N50, N75, N25, BUSCO analysis, mapping of the reads from which the assembly was generated. I see here that only the longest length is mentioned. What about all those other measurements?

Line 152. SwissProt and Tribolium casteanum genome-where it was downloaded from and which date?

Line 161-189. All this section will need to be rewritten for a better clarity. The differential gene expression is calculated using statistical tests on the mapped reads and this does not require the normalization step a priori. I am confused here on when the mapping analysis was performed, which statistical test was used to calculate the DEGs.

Line 237. Number of Project ID not listed

Round 2

Reviewer 2 Report

All my concerns have been addresses and I am suggesting the paper for publication.